# Synergy between Mecillinam and Ceftazidime/Avibactam or Avibactam against Multi-Drug-Resistant Carbapenemase-Producing *Escherichia coli* and *Klebsiella pneumoniae*

**DOI:** 10.3390/antibiotics11101280

**Published:** 2022-09-20

**Authors:** Karoline Knudsen List, Mette Kolpen, Kasper Nørskov Kragh, Godefroid Charbon, Stine Radmer, Frank Hansen, Anders Løbner-Olesen, Niels Frimodt-Møller, Frederik Boetius Hertz

**Affiliations:** 1Department of Clinical Microbiology, Rigshospitalet, DK-2100 Copenhagen, Denmark; 2Costerton Biofilm Center, Department of Immunology and Microbiology, University of Copenhagen, DK-2200 Copenhagen, Denmark; 3Department of Biology, University of Copenhagen, DK-2100 Copenhagen, Denmark; 4Statens Serum Institut, DK-2300 Copenhagen, Denmark

**Keywords:** CPE, mecillinam, ceftazidime/avibactam, multi-drug resistance

## Abstract

Background: Carbapenemase-producing *Klebsiella pneumoniae* and *Escherichia coli* have become a significant global health challenge. This has created an urgent need for new treatment modalities. We evaluated the efficacy of mecillinam in combination with either avibactam or ceftazidime/avibactam against carbapenemase-producing clinical isolates. Materials and methods: Nineteen MDR clinical isolates of *K. pneumoniae* and *E. coli* were selected for the presence of *bla*KPC, *bla*NDM, *bla*OXA or *bla*IMP based on whole-genome sequencing and phenotypic susceptibility testing. We tested the synergy between mecillinam and avibactam or ceftazidime/avibactam. We used time–kill studies in vitro and a mouse peritonitis/sepsis model to confirm the synergistic effect. We investigated avibactam’s impact on mecillinam´s affinity for penicillin-binding proteins with a Bocillin assay, and cell changes with phase-contrast and confocal laser scanning microscopy. Results: Mecillinam combined with ceftazidime/avibactam or avibactam substantially reduced MICs (from up to >256 µg/mL to <0.0016 µg/mL) for 17/18 strains. Significant log-CFU reductions were confirmed in time–kill and in vivo experiments. The Bocillin assay did not reveal changes. Conclusion: Mecillinam in combination with avibactam or ceftazidime/avibactam has a notable effect on most types of CPEs, both in vitro and in vivo. The mecillinam/avibactam combination treatment could be a new efficient antibiotic treatment against multi-drug-resistant carbapenemase-producing Gram-negative pathogens.

## 1. Introduction

The rapid spread of multi-drug-resistant Gram-negative bacteria (MDR-GNB), particularly *Escherichia coli* and *Klebsiella pneumoniae*, is of major clinical concern due to the lack of effective antibiotics, with new treatment modalities being an urgent unmet medical need [1,2,3,4]. Carbapenemases are enzymes capable of hydrolyzing almost all β-lactam antibiotics, including carbapenems. In Enterobacterales, carbapenemases are often plasmid-acquired. These plasmids commonly carry other resistance genes such as extended-spectrum β-lactamases (ESBL) and genes causing resistance to other classes of antibiotics (e.g., aminoglycosides and quinolones). Carbapenemases, such as *Klebsiella pneumoniae* carbapenemase (KPC)-type enzymes (class A), are not inhibited by older β-lactamase inhibitors such as clavulanate or tazobactam. Still, they may be inhibited by new inhibitors such as avibactam and vaborbactam. However, metallo-β-lactamases (MBLs) are not inhibited by any approved β-lactamase inhibitor (BLI), and as they hydrolyze most β-lactams, including carbapenems, they pose a substantial clinical threat. Interestingly, Marrs et al. noted that Enterobacterales with NDM-1 are commonly susceptible to mecillinam [5]. Additionally, Zykov et al. showed the effect of mecillinam against an NDM-1-producing *E. coli* strain in an experimental urinary tract infection model, and others found mecillinam-susceptible isolates among a large collection of carbapenemase-producing Enterobacterales, producing OXA-48, OXA-48-like, IMI and NDM-1 carbapenemases [6,7,8]. Thus, mecillinam may constitute a treatment option for these MDR-GNB. The purpose of our study was to evaluate the efficacy of mecillinam in combination with either avibactam or ceftazidime/avibactam against 18 carbapenemase-producing MDR clinical isolates of *K. pneumoniae* and *E. coli*, in vitro and in vivo.

## 2. Results

### 2.1. A Synergistic Effect between Mecillinam and Ceftazidime/Avibactam or Avibactam on Carbapenemase-Producing Strains

Appendix A and Table 1 show the β-lactamase genes present in the strains and the MIC determination for mecillinam, ceftazidime and ceftazidime/avibactam acquired by using the gradient test for all strains of *K. pneumoniae* and *E. coli*. As expected, all KPC carbapenemase (class A)-producing strains were susceptible to ceftazidime/avibactam, with MICs in the range of 1–3 mg/L, while all NDM (class B)-producing *K. pneumoniae* strains were resistant to ceftazidime/avibactam (>256 mg/L). Mecillinam MICs of NDM- and KPC-producing strains were in the ranges of 1–>256 mg/L and 16–>256 mg/L, respectively. All five KPC-producing *K. pneumoniae* (KP1-5) strains had mecillinam MIC > 8 mg/L, which is classified as resistant (EUCAST Clinical Breakpoint Tables v. 12.0). However, three *K. pneumoniae* strains, KP6, KP8 and KP9, that possess NDM carbapenemase had mecillinam MIC ≤ 8 mg/L. Furthermore, all six *E. coli* (EC2–7) isolates producing either NDM or OXA had mecillinam MIC ≤ 8 mg/L, while one isolate producing IMP (class B) was resistant to mecillinam (KP12). Of note, KP7 (NDM-1, OXA-48) had MICs of >256 mg/L for mecillinam, ceftazidime/avibactam and ceftazidime, respectively.

Ceftazidime/avibactam Etest strips combined with 1 or 2 mg/L mecillinam led to a ceftazidime/avibactam MIC decrease in almost all strains up to ≤0.016 mg/L, except those of KP9 (NDM-7, OXA-181), EC3 (NDM-5) and KP12 (IMP-1). The most remarkable result was KP7 (NDM-1, OXA-48), for which ceftazidime/avibactam combined with mecillinam led to the decrease in ceftazidime/avibactam MIC of up to ≤0.016 mg/L, despite the MICs for mecillinam, ceftazidime/avibactam and ceftazidime, respectively, being >256 mg/L.

Table 1 shows the susceptibility patterns of the isolates, while Table 2 shows the results of the synergistic tests of mecillinam combined with either ceftazidime/avibactam or avibactam. Interestingly, 1 or 4 mg/L avibactam combined with mecillinam Etests decreased the mecillinam MIC of KP7 to 2 and 1 mg/L, respectively. A similar decrease in the mecillinam MIC was seen in combination with 4 and 16 mg/L ceftazidime/avibactam (Table 2). All ceftazidime/avibactam-resistant *K. pneumoniae* strains showed a similar MIC decrease when mecillinam Etests were combined with avibactam or ceftazidime/avibactam (Table 2).

### 2.2. Mecillinam and Avibactam Combination Treatment in Time–Kill Assays of K. pneumoniae Indicates Synergism

All the time–kill assays showed that the combination treatment of 1 mg/l avibactam and 4 mg/L mecillinam effectively killed several *K. pneumoniae* isolates (KP2, KP5, KP6, KP7, KP8 and *K. pneumoniae* ATCC 13883) (Appendix A).

### 2.3. Mecillinam Combination Treatment with Avibactam or Ceftazidime/Avibactam In Vivo

Figure 1 and Figure 2 compare the Log CFU/mL counts in peritoneal fluids and blood at the end of the antibiotic treatment to mock treated controls at the same time point (i.e., at 5 h). The mock group depicted the increase in bacterial burden in saline-treated animals during the treatment period. Thus, the figures compared all Log CFU/mL counts at the end of the treatment period with animals sacrificed at 0 h, i.e., immediately before antibiotic treatment was initiated. The reductions in Log CFU/mL counts caused by the antibiotic treatment appeared smaller, as the difference was calculated relative to controls at baseline, instead of comparing to 5 h mock-treated controls. The efficacy of mecillinam, ceftazidime and mecillinam + ceftazidime/avibactam was evaluated as the reduction in CFU/mL in peritoneal fluid and blood collected 5 h after the initiation of the antibiotic treatment compared to Log CFU/mL counts obtained from control mice at the time of the initiation of therapy.

A reduction in Log CFU/mL was observed for the tested *E. coli* regardless of antibiotic treatment. Yet, ceftazidime/avibactam monotherapy failed to produce a log reduction in the peritoneal fluid for EC4 (NDM-5 and OXA-181) and EC5 (NDM7) (Figure 2a,b).

For all *K. pneumoniae* isolates with a KPC (KP1, KP2, KP3, KP4 and KP5) or an OXA (KP10 and KP11), ceftazidime/avibactam monotherapy was sufficient to produce a reduction in Log CFU/mL, while mecillinam failed to reduce Log CFU/mL. Of note, for KP5, there was a reduction in Log CFU/mL for all groups, including the mock-treated mice, indicating that the infection was not successful. For the *K. pneumoniae* isolate KP6 carrying the NDM carbapenemase, there was a reduction in the mean Log CFU/mL for all treatment groups, while Log CFU/mL reductions for KP7 (NDM-1 and OXA-48) were lower for mecillinam + ceftazidime/avibactam than for the other groups (Figure 3). As seen in “Appendix A from in vivo studies_log_change”, the KP7 combinational treatment with 200 mg/kg mecillinam combined with 100 mg/kg ceftazidime/avibactam in the mouse peritonitis/sepsis model led to a 3.4-log reduction (*p* = 0.001) in blood samples and 1.0-log reduction in peritoneal samples. However, this was seen for the treatment regimen containing the high doses and hourly administration only. Of note, for KP8 (NDM-5), the situation was like that mentioned above for KP5.

### 2.4. The Cell Shapes of KP7 and KP11 Are Strongly Affected by the Mecillinam and Avibactam Combination In Vitro and Ex Vivo

In vitro, bacterial suspensions were treated with increasing mecillinam and avibactam concentrations (four-fold serial dilution, 0.25–64 mg/L). At time points 3 h, 6 h and 24 h, the cell morphology was examined with phase-contrast microscopy. In addition, the combination treatment with mecillinam and avibactam was investigated, including 1 mg/L avibactam + 4 mg/L mecillinam, 4 mg/L avibactam + 16 mg/L mecillinam and 16 mg/L avibactam + 64 mg/L mecillinam. In this experiment, spherical forms of the *K. pneumoniae* strain KP7 were observed. The cell shape was more affected when avibactam and mecillinam were combined.

For KP7, high concentrations (16 mg/L and 64 mg/L) of mecillinam led to changes in cell morphology. Lower concentrations of avibactam (≥1 mg/L) were required to affect the cell shape compared to mecillinam. Interestingly, cells treated with ≥1 mg/L avibactam combined with ≥4 mg/L mecillinam transformed into spherical forms after only 3 or 6 h. However, cells treated with the corresponding concentrations of mecillinam or avibactam alone did not change after 24 h (Figure 3, Appendix A). Peritoneal fluid samples from untreated and treated mice (KP7 and KP11) were also examined ex vivo with phase-contrast microscopy. Cells also changed to spherical forms (Appendix A).

KP7 cultures treated with mecillinam and/or avibactam were stained with live–dead staining and imaged using confocal laser scanning microscopy (Figure 4, Appendix A). A green fluorescent color (syto9) stained all cells, whereas a red stain (propidium iodide) stained dead or dying cells only. The multi-resistant KP7 stain was not affected at all by the mono-treatment of either mecillinam or avibactam. Still, we saw a large decline in the surviving fraction when treated with the combination treatment (Figure 4, Appendix A).

### 2.5. Combination of Mecillinam and Avibactam Does Not Alter Their Specific Interaction with Penicillin-Binding Protein In Vitro

To investigate whether the synergy observed could be explained by an alteration of the binding affinity of mecillinam and avibactam to PBPs when the two drugs were combined, we measured the binding of a fluorescent penicillin (Bocillin FL) on PBPs extracted from an *E. coli* laboratory strain (MG1655) in the presence of avibactam and/or mecillinam. The results showed that avibactam did not affect the mecillinam PBP-binding affinity (Appendix A).

## 3. Discussion

In this study, we found a synergistic effect of the combination of ceftazidime/avibactam or avibactam alone with mecillinam against most MDR carbapenemase-producing *E. coli* and *K. pneumoniae*. In most cases, the presence of 1–4 µg of mecillinam reduced the MIC of ceftazidime/avibactam from >256 mg/L to ≤0.016 mg/L, and the presence of 1–4 mg/L of avibactam reduced mecillinam MICs from >256 mg/L to 0.25–1 mg/L. An in vitro and in vivo kill effect of both the individual antibiotics and the combination treatment was observed. We found morphological changes in the cells receiving mecillinam and avibactam, confirming the activity of the two compounds, based on binding to PBP2 in both organisms.

It is known that avibactam inhibits class A carbapenemases (e.g., KPCs) and some class D carbapenemases such as OXA-48. Avibactam is not active against NDMs [9]; however, NDM-producing *E. coli* strains have shown high susceptibility (96.5%) towards mecillinam [5,7]. A study by Marrs et al. (2014) showed lower sensitivity against mecillinam among NDM-1-producing *K. pneumoniae* strains (50%) [5]. In correlation with these findings, our study indicated that mecillinam was active against many NDM or OXA-producing strains, while ceftazidime/avibactam was active against OXA or KPC strains. The *K. pneumoniae* strain KP7 (NDM-1 and OXA-48) was resistant to mecillinam, ceftazidime/avibactam and ceftazidime alone (>256 mg/L), but mecillinam in combination with ceftazidime/avibactam or avibactam decreased the MIC to <0.016 and 1 mg/L. However, the mechanism behind this effect is unclear.

Previous studies reported that mecillinam was stable against carbapenemase-producing Enterobacterales that produce OXA-48, OXA-48-like, IMI and NDM-1 carbapenemases [5,6,8]. Yet, few of the studies reported which other β-lactamases were present in the isolates. Decreased susceptibility to β-lactams, e.g., piperacillin/tazobactam and mecillinam, could be caused by the hyperproduction of β-lactamases not considered to be ESBL, such as TEM-1 or OXA-1. Avibactam’s inhibition of other β-lactamases, while mecillinam was stable towards NDM, could be the mechanism behind the altered susceptibility we reported, as seen with aztreonam + ceftazidime/avibactam [10]. Similarly, a significant reduction in mecillinam MICs and reversion in the inoculum effect were seen in the presence of clavulanic acid [11].

The killing effect of the mecillinam/avibactam combination treatment seemed to be concentration or concentration–time-dependent. A significant reduction in colony counts in blood appeared in the KP7-infected mice treated with the combination treatment given every hour. The bacterial morphology examinations also illustrated this concentration dependency. Increasing levels of mecillinam and avibactam led to morphological changes and increased cell size of KP7.

Previous studies showed a strong binding of mecillinam and avibactam to *E. coli* and *K. pneumoniae* PBP2 [12,13,14]. We also found that mecillinam and avibactam had a high affinity for PBP2 (IC50: 0.3 and 0.2 mg/L, respectively). However, our Bocillin FL assay did not show if the synergistic effect between mecillinam and avibactam was due to a changed binding affinity of mecillinam and avibactam to *E. coli* PBPs. Of note, Erlinda et al. found that avibactam could interact with PBP2, which led to bactericidal interactions with human cathelicidin antimicrobial peptide LL-37, an antibacterial peptide part of innate immunity [12,15]. The observation of no visible decrease in mecillinam IC50 for any of the PBPs when mecillinam was combined with an avibactam concentration almost seven-fold higher than the avibactam IC50 for PBP2 could indicate that the mechanism behind the synergistic effect of the combination treatment, in vitro and in vivo, was that both mecillinam and avibactam could bind to PBP2, while avibactam could partly inhibit non-NDM β-lactamases. Further studies are needed to clarify why the Bocillin FL-binding did not change when avibactam, at a concentration seven-fold higher than IC50, was combined with β-lactams. Notably, currently, there is insufficient evidence for the use of mecillinam for the management of systemic infections with MDR carbapenemase-producing *E. coli* and *K. pneumoniae*.

Considering the limitations of our study, it should be noted that accurate mecillinam MIC determination by using gradient tests can be challenging, as previously shown by Fuchs et al. [6]. Further, the gradient strips used a fixed concentration of avibactam, so the result of the MIC with ceftazidime might have been more influenced by the avibactam concentration.

In conclusion, the use of the combination of mecillinam and avibactam must be investigated further. Yet, there is obvious potential for the treatment of urinary tract infections.

## 4. Materials and Methods

### 4.1. Bacterial Isolates

A collection of 18 varying phylogenetic clinical isolates of *E. coli* and *K. pneumoniae,* all phenotypically resistant to meropenem and carrying at least one carbapenemase gene, identified with whole-genome sequencing (WGS), was investigated (Appendix A). *E. coli* MG1655, *E. coli* ATCC25922 and *K. pneumoniae* ATCC13883 were included as controls. All strains were stored at −80 °C until use.

### 4.2. Antibiotics

Mecillinam (Karo Pharma, Stockholm, Sweden), ceftazidime/avibactam (Pfizer, Kent, UK), avibactam (MedChemExpress, Monmouth Junction, NJ, USA) and ceftazidime (Fresenius Kabi, Singapore) were dissolved in MilliQ water. The fluorescent penicillin Bocillin FL (Invitrogen, Eugen, OR, USA) was used to label penicillin-binding proteins (PBPs) in the Bocillin FL assay. In addition, cefsulodin (Glentham Life Sciences Ltd., Corsham, UK), aztreonam (Sanofi-aventis, Paris, France) and cefoxitin (Novopharm Limited, Toronto, ON, Canada) were used as control antibiotics in the Bocillin FL assay.

### 4.3. Broth and Growth Conditions

We chose the combination of mecillinam and ceftazidime/avibactam based on an initial evaluation of multiple antibiotic combinations, tested in the chequerboard assay and performed in Mueller–Hinton broth [16]. For the susceptibility test using gradient test screening, each isolate was suspended in 0.9% NaCl until reaching a density of 0.5 McFarland. Gradient test strip studies were performed on Mueller-Hinton II (MHII) agar plates with and without antibiotics, which were produced by the University of Copenhagen, Panum. Plates were incubated at 35 °C for 16–20 h. The start inoculum of the time–kill assays and confocal laser scanning microscopy experiments was 10^5^ colony-forming units CFU/mL, while a start inoculum of 10^7^ CFU//mL was used for the phase-contrast microscopy experiments. All bacterial isolates were grown in 10 mL cation-adjusted MHII broth (17.5 g/L of acid hydrolysate of casein, 3 g/L of beef extract and 1.5 g/L starch) at 37 °C, 150 rpm. A growth control was included in all experiments. All test tubes of the time–kill assays were incubated for 6 h. Of note, we used 6 h incubation, since it was previously been found that mecillinam degradation can occur rapidly, with a half-life as short as 2 h in medium at 37 °C and pH 7.4 [17]. For CFU counts, bacterial suspensions were plated onto solid Gram-negative selective lactose agar plates (bromothymol agar plates based on modified Conradi-Drigalski medium containing 10 g/L detergent, 1 g/L Na_2_S_2_O_3_ H_2_O, 0.1 g/L bromothymol blue, 9 g/L lactose and 0.4 g/L glucose, pH 8.0, SSI Diagnostica). Colonies were counted after 16–20 h incubation at 37 °C.

### 4.4. Whole-Genome Sequencing

Genomic DNA was extracted using the DNeasy Blood and Tissue Kit (Qiagen, Hilden, Germany), followed by library preparation with the Nextera XT DNA Library Preparation Kit (Illumina, San Diego, CA, USA) and short-read whole-genome sequencing using 251 bp paired-end MiSeq sequencing (Illumina), according to the manufacturer’s instructions. Raw data were assembled into draft genomes using Skesa v2.2 [18]. ENA Accession numbers for raw Illumina reads for the study isolates can be found in the Appendix A.

### 4.5. MIC Determination

Minimal inhibitory concentrations (MICs) of mecillinam, ceftazidime/avibactam and ceftazidime were determined by performing a gradient test. Ceftazidime/avibactam and ceftazidime Etest strips were obtained from BioMérieux, Ballerup, Denmark. Mecillinam MIC test strips were from Liofilchem (Roseto degli Abruzzi, Italy).

Bacterial susceptibility to the combinations mecillinam + ceftazidime/avibactam, mecillinam + ceftazidime and mecillinam + avibactam was tested as follows: ceftazidime/avibactam and ceftazidime strips were placed on MHII agar plates containing 0.5, 1 or 2 mg/L mecillinam, while mecillinam strips were placed on MHII agar plates with 1 or 4 mg/L avibactam and 4 or 16 mg/L ceftazidime/avibactam. Breakpoints for Enterobacterales were set for mecillinam (S ≤ 8 mg/L, R > 8 mg/L), ceftazidime (S ≤ 1 mg/L, R > 4 mg/L) and ceftazidime/avibactam (S ≤ 8 mg/L, R > 8 mg/L) (EUCAST Clinical Breakpoint Tables v. 12.0).

### 4.6. Binding of Mecillinam and Avibactam to Penicillin-Binding Proteins (PBPs) Measured with Bocillin FL Assay

The binding of mecillinam and avibactam to PBPs was examined through a competition assay with the fluorescent penicillin Bocillin FL. For details on the purification of *E. coli* MG1655 membrane proteins, see the Appendix A. In total, 30 µg of *E. coli* MG1655 membrane protein was mixed with 2-fold dilutions (0.0625–32 mg/L) of mecillinam and avibactam and incubated at 35 °C for 30 min. Reaction mixtures containing 3 mg/L cefsulodin, 2 mg/L aztreonam and 2 mg/L cefoxitin were prepared to localize PBP1, PBP3, PBP4, PBP5 and PBP6. An assay containing reaction mixtures with a fixed avibactam concentration of 2 mg/L and 2-fold dilutions (0.0625–32 mg/L) of mecillinam was used. After incubation with antibiotics, Bocillin FL was added to reaction mixtures for further incubation at 35 °C for 40 min. To denature proteins, an SDS-PAGE-loading buffer with β-mercaptoethanol was added to each mixture and incubated at 95 °C for 5 min before the SDS-PAGE loading. An SDS-PAGE analysis was performed using 4–12% NuPAGE Bis-Tris Midi Gels (Thermo Fischer Scientific, Waltham, MA, USA). After complete protein separation, the gel was rinsed in water three times and for 30 min in a fixing solution containing 45% methanol and 7% acetic acid. The gel was scanned using a Typhoon 7000 scanner with excitation of 473 nm and emission of 520 nm. The ImageJ software was used to analyze the Bocillin FL signal of the gel, leading to the determination of integrated densities and percent values. The background signal of the gel image was subtracted before the band analysis. Relative Bocillin FL binding was determined by dividing each PBP percent value with the standard percent value of the reaction mixture without antibiotics, which indicated 100% Bocillin FL binding.

### 4.7. Phase-Contrast Microscopy

A Nikon Eclipse Ti was used for all phase-contrast microscopy examinations. Samples were spotted onto 1.5% agarose-MHII pads before imaging. Nikon’s imaging software, NIS-Elements Advanced Research Version 4.40, was used to process phase-contrast microscope images.

### 4.8. Confocal Laser Scanning Microscopy

Bacterial cells were examined by using confocal laser scanning microscopy. Samples were diluted 1:10 in saline containing 3 mM Syto9 and 0.3 mM of propidium iodide (PI) (live/dead staining) (Thermo Fisher Scientific, USA). Samples were incubated for 15 min in the absence of visible light, before 30 µL was applied to the wells of a VI µ-slide microscopy slide (Ibidi, Gräfelfing, Germany). Samples were imaged using a 488 nm laser for excitation, a 495–550 nm emission filter for Syto9, a 561 nm laser and a 595–650 nm emission filter for Pl. All samples were evaluated in three random positions within the wells. The images were processed in Imaris 9.2 (Bitplane, Zürich, Switzerland) for qualitative 3D projections and biomass quantitation via the MeasuremePro addon in Imaris. Syto9-stained biomass and PI-stained biomass were quantified in parallel to give a total biomass and indicate the viability of the biomass.

### 4.9. Single and Combination Antibiotic Treatment In Vivo in the Mouse Peritonitis/Sepsis Model

All in vivo experiments were conducted according to the permission of the Danish Animal Ethical Council under license 2017–15-0201–01274. We performed in vivo mouse experiments for all isolates where in vitro susceptibility testing indicated an in vivo effect (EC4, EC5, EC6, EC7, KP1, KP2, KP3, KP4, KP5, KP6, KP7, KP8, KP10 and KP11). Each experiment included 36 outbred female NMRI mice (seven weeks old, weighing 26–30 g) (Taconic, Hürth, Germany) divided into five groups: four mice in a control group for determining the baseline infection load at the start of treatment, eight mice per treatment group and eight mice treated with saline (0.9% NaCl) as the control. At t = 0, mice were inoculated intraperitoneally with 0.5 mL of bacterial suspension containing 10^7^ CFU/mL and 5% (*w*/*v*) porcine mucin (Sigma-Aldrich, St. Louis, MO, USA). Blood and peritoneal fluid were sampled from four mice from the control group after one hour for determining the level of infection before treatment. Blood from the submandibular vein was collected. After euthanizing the mice through cervical dislocation, a peritoneal wash was performed by injecting 2 mL of sterile saline, i.e., performing a gentle massage of the abdomen for 1 min and then opening the abdomen aseptically for the sampling of the peritoneal fluid with a pipette. During the study, at t = 1 h, t = 3 h and t = 5 h, the mice were observed, and their clinical scores were noted according to the grades that can be found in the Supplementary data. The remaining mice in the treatment groups were injected (s.c. in the thigh) with 0.2 mL of antibiotic X, antibiotic Y, X and Y or saline. The doses used for ceftazidime/avibactam and mecillinam would result in serum drug concentrations in mice similar to concentrations in humans on standard doses [19,20]. The ENA accession numbers for raw Illumina reads for the study isolates and source of isolates can be found in the Appendix A.
Setup 1: 100 mg kg^−1^ ceftazidime/avibactam alone, 50 mg kg^−1^ mecillinam alone, 100 mg kg^−1^ ceftazidime/avibactam and 50 mg kg^−1^ mecillinam together or saline [19,20].Setup 2: 200 mg kg^−1^ ceftazidime/avibactam alone, 200 mg kg^−1^ mecillinam alone, 200 mg kg^−1^ ceftazidime/avibactam and 200 mg kg^−1^ mecillinam together or saline [19,20].Setup 3: 200 mg kg^−1^ avibactam alone, 100 mg kg^−1^ mecillinam alone, 200 mg kg^−1^ avibactam and 100 mg kg^−1^ mecillinam together or saline.

Mice treated with two antibiotics were each injected into separate thighs. At t = 3 h, after 2 h of treatment, four mice from each group were removed, and blood and peritoneal fluid samples were taken as described above. For tests in mice involving antibiotics with a low half-life (<1 h), these antibiotics were boosted with a second injection at t = 3 h (setup 1 and 2) or at t = 2 h, t = 3 h and t = 4 h (setup 3). At t = 5 h, the last four mice in each group were removed, and samples were taken as described above. All samples were stored at 4 °C and were processed within 1 h of sampling. Samples were 10-fold serially diluted in saline before being plated onto solid lactose agar plates (‘blue plates’) for CFU counts. Besides CFU, data were presented as log reductions as per the untreated saline group.

### 4.10. Statistical Methods

For comparisons of colony counts in vitro or in vivo, we used nonparametric statistics, either Fisher´s exact test or the Kruskal–Wallis test. For the determination of IC50 for penicillin-binding protein binding, we used a four-parameter dose–response model in the GraphPad Prism Version 9 software. *p* < 0.05 was considered significant.

## 5. Conclusions

In conclusion, the combination of ceftazidime/avibactam with mecillinam decreased Log CFU/mL in peritoneum against all 18 strains, irrespective of all narrow and broad-spectrum β-lactamases present in the pathogens. Therefore, the combination of mecillinam–ceftazidime/avibactam or mecillinam–avibactam combination treatments could be a new efficient empiric antibiotic treatment to fight infections caused by MDR carbapenemase-producing Gram-negative pathogens. The mechanism behind the observed synergistic effect between mecillinam and avibactam should be investigated further.

## Figures and Tables

**Figure 1 antibiotics-11-01280-f001:**
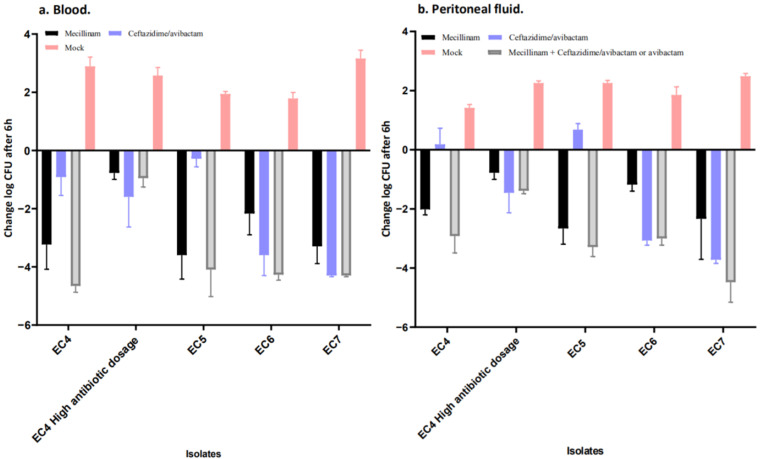
(**a**,**b**). In vivo results of treatment regimens for the included *E. coli* isolates. The standard doses were 100 mg kg^−1^ ceftazidime/avibactam alone, 50 mg kg^−1^ mecillinam alone, 100 mg kg^−1^ ceftazidime/avibactam and 50 mg kg^−1^ mecillinam together or saline administered s.c. at t = 1 h and t = 3 h. For strain EC4, a higher dose was also used, i.e., 200 mg kg^−1^ ceftazidime/avibactam alone, 200 mg kg^−1^ mecillinam alone, 200 mg kg^−1^ ceftazidime/avibactam and 200 mg kg^−1^ mecillinam together or saline administered at t = 1 h, t = 2 h, t = 3 h and t = 4 h, indicated as “High antibiotic dosage”. Data are presented as changes in log CFU mL−1 in either blood or peritoneal fluid at t = 5 h. The figure compares CFU counts in peritoneal fluid and blood at the end of antibiotic treatment to mock-treated controls at the same time points. Thus, the figure compares all CFU counts at the end of the treatment period with animals sacrificed at 0 h, i.e., immediately before antibiotic treatment was initiated. The result of this new depiction is that reductions in CFU counts caused by antibiotic treatment appeared smaller as the difference was calculated relative to controls at baseline with lower counts instead of compared to 5 h mock-treated controls. The “mock” group depicted the increase in bacterial burden in mock-treated animals during the treatment period.

**Figure 2 antibiotics-11-01280-f002:**
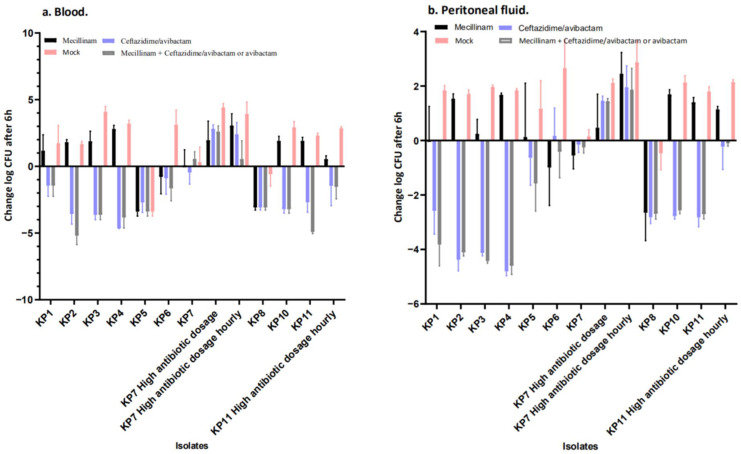
(**a**,**b**). In vivo results of treatment regimens for the included *K. pneumoniae* isolates. The standard doses were 100 mg kg^−1^ ceftazidime/avibactam alone, 50 mg kg^−1^ mecillinam alone, 100 mg kg^−1^ ceftazidime/avibactam and 50 mg kg^−1^ mecillinam together or saline administered s.c. at t = 1 h and t = 3 h. For strain KP7, two higher dose regimens were used, i.e., 200 mg kg^−1^ ceftazidime/avibactam alone, 200 mg kg^−1^ mecillinam alone, 200 mg kg^−1^ ceftazidime/avibactam and 200 mg kg^−1^ mecillinam together or saline administered at t = 1 h and t = 3 h (marked in the figure with “high antibiotic dosage”), and the same higher dose administered hourly, i.e., t = 1 h, t = 2 h, t = 3 h and t = 4 h (marked in the figure with “high antibiotic dosage hourly”). This latter dose was also used for strain KP11. Data are presented as changes in log CFU mL^−1^ in either blood or peritoneal fluid t = 5 h. The figure compares CFU counts in peritoneal fluid and blood at the end of antibiotic treatment to mock-treated controls at the same time points. Thus, the figure compares all CFU counts at the end of the treatment period with animals sacrificed at 0 h, i.e., immediately before antibiotic treatment was initiated. The result of this new depiction is that reductions in CFU counts caused by antibiotic treatment appeared smaller as the difference was calculated relative to controls at baseline with lower counts instead of compared to 5 h mock-treated controls. The “mock” group depicted the increase in bacterial burden in mock-treated animals during the treatment period.

**Figure 3 antibiotics-11-01280-f003:**
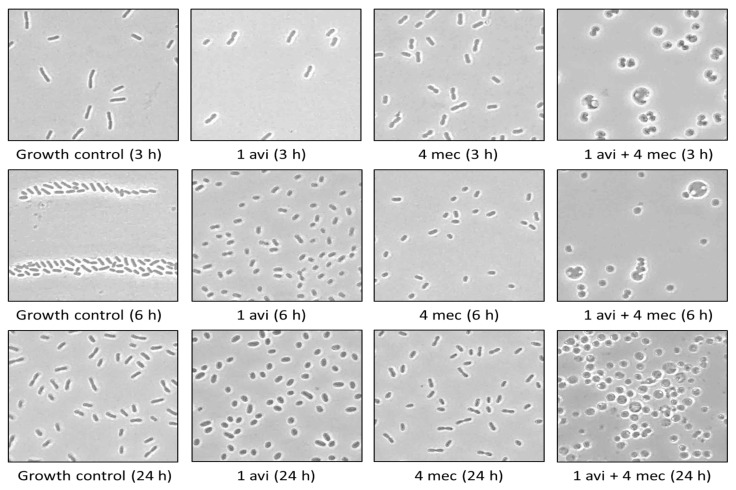
Spherical structures of the *K. pneumoniae* strain KP7 observed in vitro. Phase-contrast microscopy (1000× magnification) of KP7 cells in four treatment groups, including growth control, 1 mg/L avibactam (1 AVI), 4 mg/L mecillinam (4 MEC) and 1 mg/L avibactam combined with 4 mg/L mecillinam (1 AVI + 4 MEC). Images are from time points 3, 6 and 24 h.

**Figure 4 antibiotics-11-01280-f004:**
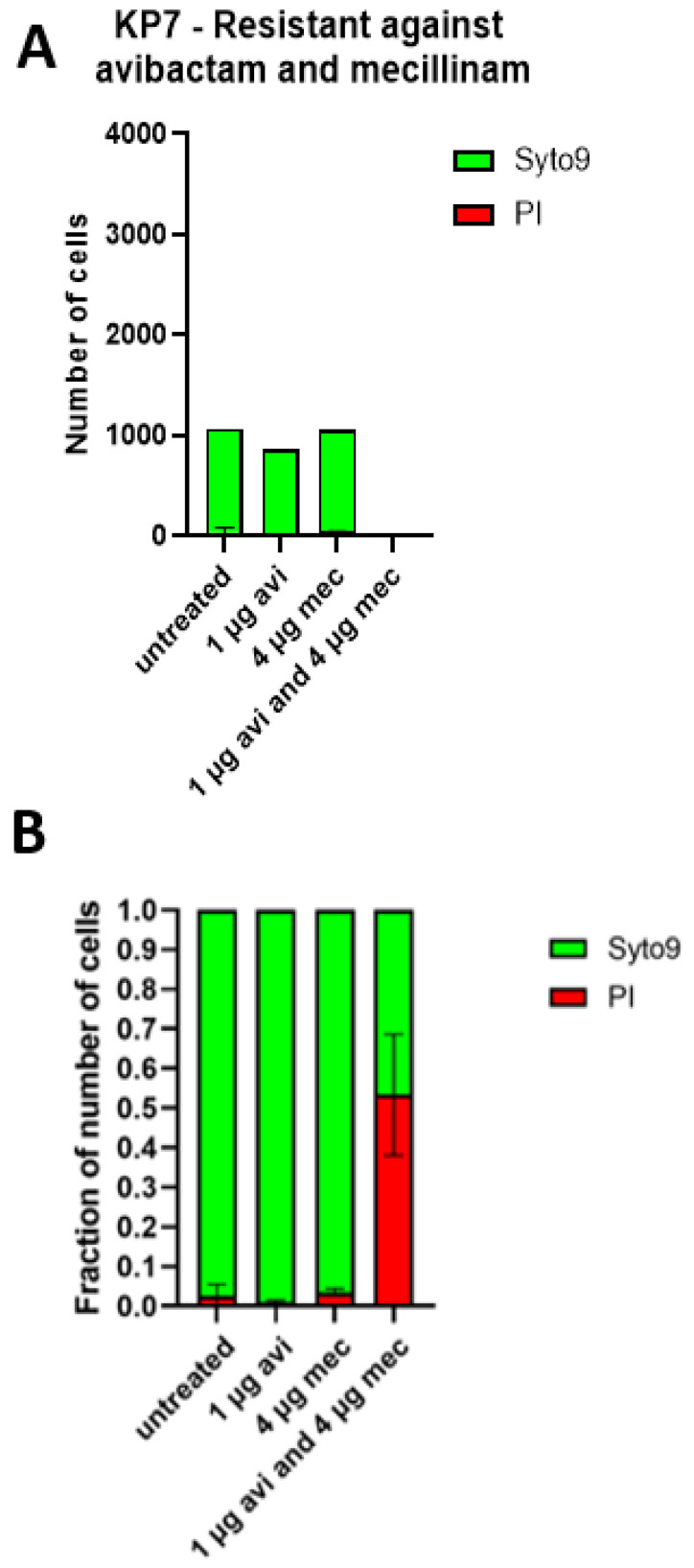
Confocal laser scanning microscopy. Quantified data showed the average number of cells found in confocal laser scanning micrographs of cultures treated either with 4 µg mL^−1^ mecillinam, 1 µg mL^−1^ avibactam or untreated. (**A**) The number of cells of the syto9 (live) and the PI (dead)-stained populations was quantified using Imaris 9.5 with Measure Pro. KP7 cells were resistant against avibactam and mecillinam. (**B**) A fraction of live and dead cells is shown based on the total number of cells in each sample.

**Table 1 antibiotics-11-01280-t001:** Susceptibility patterns of the included isolates as well as results of synergistic tests of mecillinam combined with ceftazidime/avibactam.

Strains	Carbapenemase	MIC Mecillinam (mg/L)	MIC Ceftazidime/Avibactam (mg/L)	MIC Ceftazidime/Avibactam (mg/L) Performed on Agar Containing Mecillinam			MIC Ceftazidime (mg/L)	MIC Ceftazidime (mg/L) Performed on Agar Containing Mecillinam
				Mecillinam: 0.5 mg/L	Mecillinam: 1 mg/L	Mecillinam: 2 mg/L		Mecillinam: 2 mg/L
**KP1**	KPC-2	16	1	0.032	<0.016	<0.016	8	8
**KP2**	KPC-3	>256	4	0.064	<0.016	<0.016	>256	>256
**KP3**	KPC-3	32	1	<0.016	<0.016	<0.016	>256	>256
**KP4**	KPC-3	>256	2	2	0.064	<0.016	>256	>256
**KP5**	KPC-3	>256	2	0.016	<0.016	<0.016	>256	>256
**KP6**	NDM-1	2	>256	>256	<0.016	<0.016	>256	>256
**KP7**	NDM-1, OXA-48	>256	>256	>256	<0.016	<0.016	>256	>256
**KP8**	NDM-5	2	>256	>256	>256	<0.016	>256	>256
**KP9**	NDM-7, OXA-181	8	>256	>256	>256	>256	>256	>256
**KP10**	OXA-232	16	1	0.25	<0.016	<0.016	32	32
**KP11**	OXA-436	4	0.5	0.032	<0.016	<0.016	64	32
**KP12**	IMP-1	>256	32	32	32	32	>256	>256
**ATCC 13883**		>256	0.25	0.032	0.016	0.016	0.25	0.016
**EC2**	NDM-1	1	>256	NG	NG	NG	>256	NG
**EC3**	NDM-5	8	>256	>256	>256	>256	>256	>256
**EC4**	NDM-5, OXA-181	4	>256	>256	>256	<0.016	>256	>256
**EC5**	NDM-7	4	>256	>256	<0.016	<0.016	>256	>256
**EC6**	OXA-244	2	0.5	<0.016	<0.016	<0.016	2	2
**EC7**	OXA-48	2	0.25	<0.016	<0.016	<0.016	0.25	0.25
**ATCC 25922**		0.064	0.125	NG	NG	NG	0.125	NG

EC: *Escherichia coli*; KP: *Klebsiella pneumoniae*; KPC: *Klebsiella pneumoniae* carbapenemase; IMP: imipenem metallo-beta-lactamase; NDM: New Delhi metallo-beta-lactamase; OXA: oxacillinase; NG: no growth on MHII agar plate.

**Table 2 antibiotics-11-01280-t002:** Results of synergistic tests of mecillinam combined with ceftazidime/avibactam or avibactam. We showed the isolates chosen for further analyses based on results from Table 1.

Strain	Carbapenemase	MIC Mecillinam (mg/L)	MIC Ceftazidime/Avibactam (mg/L)	Mecillinam MIC (mg/L) Performed on Agar Containing Avibactam		Mecillinam (mg/L) Performed on Agar Containing Ceftazidime/Avibactam	
				Avibactam: 1 mg/L	Avibactam: 4 mg/L	Ceftazidime/Avibactam: 4 mg/L	Ceftazidime/Avibactam: 16 mg/L
KP2	KPC-3	>256	4	1	0.25	1	NG
KP5	KPC-3	>256	2	1	0.25	2	NG
KP6	NDM-1	2	>256	1	0.5	1	1
KP7	NDM-1, OXA-48	>256	>256	2	1	2	2
KP8	NDM-5	2	>256	1	0.5	1	1
ATCC 13883		>256	0.25	8	8	NG	NG
EC4	NDM-5, OXA-181	4	>256	1	0.25	2	1
EC5	NDM-7	4	>256	1	0.5	1	1
ATCC 25922		0.064	0.125	0.064	0.016	NG	NG

EC: *Escherichia coli*; KP: *Klebsiella pneumoniae*; KPC: *Klebsiella pneumoniae* carbapenemase; IMP: imipenem metallo-beta-lactamase; NDM: New Delhi metallo-beta-lactamase; OXA: oxacillinase; NG: no growth on MHII agar plate.

## Data Availability

Data are presented in full in the manuscript or in the Appendix A. Raw Illumina reads are available at the European Nucleotide Archive.

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
