# Peer review of "Synergy between Mecillinam and Ceftazidime/Avibactam or Avibactam against Multi-Drug-Resistant Carbapenemase-Producing Escherichia coli and Klebsiella pneumoniae"

_antibiotics, 2022, doi:10.3390/antibiotics11101280_

Round 1

Reviewer 1 Report

Results

Line 106: Klebsiella pneumoniae should be written in italics

Material and Methods

Line 334-335: „…Plates were incubated at 37°C for 16-20 h”.

Line 342 and 347: „…37°C…”

Incubation to assess antibiotic susceptibility should be carried out at 35±1°C for 18±2h. Why did the authors use a different temperature?

Line 336-337: „….were 105 colony-forming units CFU/mL, while a start inoculum of 107 CFU//mL was used for the phase-contrast microscopy experiments”.

Shouldn't it be as follows?: „….were 105 colony-forming units CFU/mL, while a start inoculum of 107 CFU//mL was used for the phase-contrast microscopy experiments”.

Line 424: „…107 CFU/ml” should be „…107CFU/ml”

Line 451: „…at 4oC..” should be „…at 4°C..”

Supplementary material

Table S1:

If other beta-lactamases then no “bla” in front of the enzyme name

Author Response

Line 106: Klebsiella pneumoniae should be written in italics

-This has been corrected.

Material and Methods

Line 334-335: „…Plates were incubated at 37°C for 16-20 h”.

Line 342 and 347: „…37°C…”

Incubation to assess antibiotic susceptibility should be carried out at 35±1°C for 18±2h. Why did the authors use a different temperature?

-Thank you for your comments, it was indeed at 35°C and not  37°C as previously written. This is now corrected.

Line 336-337: „….were 105 colony-forming units CFU/mL, while a start inoculum of 107 CFU//mL was used for the phase-contrast microscopy experiments”.

Shouldn't it be as follows?: „….were 105 colony-forming units CFU/mL, while a start inoculum of 107 CFU//mL was used for the phase-contrast microscopy experiments”.

-This has been corrected.

Line 424: „…107 CFU/ml” should be „…107CFU/ml”

-This has been corrected.

Line 451: „…at 4oC..” should be „…at 4°C..”

-This has been corrected.

Supplementary material

Table S1:

If other beta-lactamases then no “bla” in front of the enzyme name

-This has been corrected.

Reviewer 2 Report

The authors present a very well written and presented research article on synergy between mecillinam and ceftazidime/avibactam or avibactam against multi-drug resistant Escherichia coli and Klebsiella pneumoniae. Whilst the number of isolates included is small (n=19) they are a well chosen selection of CPEs and the authors go on to conduct well considered and thought out susceptibility and effect studies.

It is a pity that the Bocillin assay did not show if the synergistic effect between mecillinam and avibactam is due to a changed affinity to E. coli PBPs but the authors go on to discuss this adequately with a suggestion for further research into mechanism of action.

The whole genome sequencing supplement is a valuable contribution as it shows the other beta-lactamases present in the isolates. This is often overlooked in many studies.

The would suggest that he authors check references 6, 8, 9, 10 and 16 for completeness and accuracy.

Author Response

The would suggest that he authors check references 6, 8, 9, 10 and 16 for completeness and accuracy.

Thank you for your observations, reference 9 was moved up, 10 was removed and former reference 13  (now 12) was added in the same sentence as former reference 16 (now 15).

Reviewer 3 Report

The paper is excellent I thoroughly enjoyed reading it. I am happy to recommend acceptance.  I only have minor comments that should be addresses

Minor Comments

Why was the study limited to K. pneumoniae and E.coli?

Please give more information (if available) on the clinical strains were they wound isolates, UTI isolates, Patient screenings isolates etc.

In lines 336, 337 and 423 presumably it should be 105  (not 105) colony-forming units CFU/mL, and 107 (not 107) CFU//mL?

Author Response

Why was the study limited to K. pneumoniae and E.coli?

We focus on K. pneumoniae and E. coli since they are the two major causes of carbapenem producing Gram-negative bacterial infections.

Please give more information (if available) on the clinical strains were they wound isolates, UTI isolates, Patient screenings isolates etc.

This has been added in Supplemental table 3.

In lines 336, 337 and 423 presumably it should be 105  (not 105) colony-forming units CFU/mL, and 107 (not 107) CFU//mL?

-This has been corrected.